

# Hydrogen dynamics in soil organic matter as determined by $^{13}$C and $^{2}$H labeling experiments

Alexia Paul[1], Christine Hatté[2], Lucie Pastor[3], Yves Thiry[4], Françoise Siclet[5], Jérôme Balesdent[1]

[1] Aix-Marseille Universite, CNRS, College de France, IRD, INRA, CEREGE UM34, 13545 Aix-en-Provence, France

[2] LSCE, UMR CEA-CNRS-UVSQ 8212, Domaine du CNRS, F-91198 Gif-sur-Yvette

[3] IFREMER/Centre de Brest, Département REM/EEP/LEP, CS 10070, 29280 Plouzané-France

[4] Andra, Research and Development Division, Parc de la Croix Blanche, 1/7 rue Jean Monnet, 92298 Châtenay-Malabry Cedex, France

[5] EDF R&D, LNHE, 6 quai Watier, 78400 Chatou, France

*Correspondence to* Alexia Paul (alexia.paul@aix.inra.fr) and Jérôme Balesdent (jerome.balesdent@aix.inra.fr)

Abstract: Understanding hydrogen dynamics in soil organic matter is important to predict the fate of $^{3}$H in terrestrial environments. One way to resolve hydrogen fate and to point out processes is to examine the isotopic signature of the element in soil. However, non-exchangeable hydrogen isotopic signal in soil is complex and depends on the fate of organic compounds and microbial biosyntheses that incorporate water-derived hydrogen.
5   To decipher this complex system and to understand the close link between hydrogen and carbon cycles, we followed labeled hydrogen and labeled carbon all along natural-like soil incubations. We performed incubation experiments with three labeling conditions: 1- $^{13}$C$^{2}$H double-labeled molecules in the presence of $^{1}$H$_2$O, 2- $^{13}$C-labeled molecules in the presence of $^{2}$H$_2$O, 3- no molecule addition in the presence of $^{2}$H$_2$O. The preservation of substrate-derived hydrogen after one year of incubation (ca. 5% in most cases) was lower than the preservation
10   of substrate-derived carbon (30% in average). We highlighted that 70% of the C-H bonds are broken during the degradation of the molecule which permits the exchange with water hydrogen. Added molecules are used more for trophic resources. The isotopic composition of the non-exchangeable hydrogen was mainly driven by the incorporation of water hydrogen during microbial biosynthesis. It is linearly correlated with the amount of carbon that is degraded in the soil. The quantitative incorporation of water hydrogen in bulk material and lipids
15   demonstrates that non-exchangeable hydrogen exists in both organic and mineral-bound forms. The proportion of the latter depends on soil type and minerals. This experiment quantified the processes affecting the isotopic composition of non-exchangeable hydrogen, and the results can be used to predict the fate of tritium in the ecosystem or the water deuterium signature in organic matter.

1   Introduction

 Our knowledge of the nature of soil organic matter (SOM) has made great progress in recent decades: it is now considered to be a continuum of progressively decomposing organic compounds (Lehmann and Kleber, 2015), composed of all the components of living material such as glucides, peptides, lipids, organic acids and phenolic compounds (Kelleher and Simpson, 2006). Small molecules are associated with each other in supramolecular
structures or with mineral particles by weak bonds, including H-bonds (Sutton and Sposito, 2005; Lehmann and



Kleber, 2015). Their lifetime in soils is controlled more by sorption or protection than from intrinsic chemical recalcitrance to biodegradation (Schmidt et al., 2011; Basile-Doelsch et al., 2015) with the exception of pyrolytic products. Highly degradable compounds, such as glucides and peptides, contribute to the oldest SOM components, and biomarkers tend to indicate that old SOM was derived more from microbial-derived products

than from plant-derived molecules as a result of the mineral protection processes (Rafter and Stout, 1970; Derrien et al., 2007; Bol et al., 2009). Carbon dynamics in this continuum have been widely studied using the natural $^{14}C/^{12}C$ and $^{13}C/^{12}C$ (Rafter and Stout, 1970; Balesdent et al., 1987) ratios and also through labeling experiments (Jenkinson, 1965; Murayama, 1988; Derrien et al., 2004). The results of these experiments have highlighted the different turnover of soil organic pools. Plant material is rapidly decomposed into microbial

biomass, and a small portion of both can be protected from biodegradation for decades to centuries, representing the main part of SOM. One part of this organic matter remains stabilized for millennia, especially in deep soil horizons. It is therefore expected that the non-exchangeable hydrogen (NEH) dynamics, bound to carbon in soil, will be controlled by the same processes: organic carbon inherited from vegetation, biodegradation, microbial biosyntheses and stabilization.

Hydrogen has various molecular positions in soil. It can be organic or inorganic and non-exchangeable or exchangeable with available hydrogen. The abiotic exchange of organic hydrogen depends on the strength of the bond and the energy required for exchange (Belot, 1986; Schimmelmann, 1991; Ciffroy et al., 2006; Sauer et al., 2009). Bound to N, O and S, hydrogen is usually exchangeable with ambient water and water vapor (Schimmelmann, 1991; Wassenaar and Hobson, 2000). However, hydrogen bound to carbon is considered to be

stable and non-exchangeable due to the strong covalent bonds (Baumgärtner and Donhaerl, 2004; Diabaté and Strack, 1997; Kim et al., 2013). At the ecosystem scale, H bound to C is not exchangeable (Sessions et al., 2004). Hydrogen can also interfere with clay minerals. Interlayer water exchanges with free water within a few hours and is removed after drying (Savin and Epstein, 1970). However, the structural water and the hydroxyl hydrogen of clay are non-exchangeable at room temperature (Savin and Epstein, 1970).

The natural $^{2}H/^{1}H$ ratio of plants and sediments has been used as a proxy to reconstruct past climate and paleoenvironmental conditions, such as temperature, water use efficiency (Epstein et al., 1976; Sessions et al., 2004; Zech et al., 2014; Tuthorn et al., 2015) The isotopic composition of the NEH preserves the initial composition of the plant and registers the rain isotopic composition (Sessions et al., 2004; Schimmelmann et al., 2006; Ruppenthal et al., 2010). The $\delta^{2}H$ of water and exchangeable hydrogen is not stable. Whereas soil organic

carbon and nitrogen cycles have been extensively studied, soil organic hydrogen and its recycling in the environment remain poorly understood due to its complex behavior. The total bulk soil, composed of a mixture of non-exchangeable and exchangeable, organic and inorganic hydrogen, makes the hydrogen isotopic composition hard to determine. The composition and exchanges between these pools can be of great importance when modelling, for instance, tritium fate in the environment. Models used to predict the fate and behavior of

tritium in the environment often simplify processes linked to the formation and degradation of organic bound tritium in plants and it is not often taken into account in soils processes. But organically bound tritium of terrestrial origin has been unexpectedly observed in rivers at higher concentrations than the concentration of tritium in water because of the historic nuclear testing (Gontier and Siclet, 2011; Kim et al., 2012; Eyrolle-Boyer et al., 2014). And the concentration of organic tritium in soil can often be higher than the concentration of water

tritium in soil (Thompson et al., 2015) due to its longer residence time. There is thus great concern about



organically bound tritium in ecosystems, which may result from tritium release in the environment. Consequently, there is a need to assess the fate and residence time of organically bound tritium. Therefore, it is necessary to quantify the preservation of organic hydrogen from vegetation, the accumulation of hydrogen from water in the soil and the processes involved in organic matter decomposition and mineralization.

To decipher and quantify the preservation of the organic material and the microbial biosyntheses incorporating water-derived hydrogen, we designed incubation experiments with labeled compounds by assuming that the non-exchangeable hydrogen dynamics are controlled by the carbon dynamics in the soil organic matter. Three scenarios were addressed:

1- $^{13}C^2H$-double labeled molecules in the presence of $^1H_2O$, 2- $^{13}C$-labeled molecules in the presence of $^2H_2O$, 3-

no molecule addition in the presence of $^2H_2O$. The $^{13}C$ and $^2H$ bulk soil isotopic compositions were analyzed at different times to quantify the processes involved. The isotopic composition of lipids was also analyzed as an indicator of organically bound NEH.

The medium-term $^{13}C$ and $^2H$ labeling experiments were conducted on different types of soil (clayey leptosol, cambisol and podzol) from 0 to 1 year to highlight and quantify the processes affecting hydrogen based on the

carbon dynamics in the soil organic matter and their dependence on the soil properties.

## 2 Materials and Methods

### 2.1 Soil sampling


Three soils with contrasting physical and chemical properties were selected for this study (Table 1):

Cambisol: the 0-25 cm surface layer of a cambisol was sampled from an INRA long-term field experiment in the Parc du Château de Versailles, France in March 2014. This soil is a neutral eutric cambisol with a clay composition of 17%, 33% of sand, 50% of silt and a nil carbonate content (Dignac et al., 2005). The plant cover

is wheat. After each harvest, wheat residues were returned to the soil, and the first 25 cm were ploughed each fall.

Podzol: the 0-25 cm surface layer of a podzol was sampled from an INRA field experiment in Pierroton (close to Bordeaux, SW France) in May 2014. The lands cover of les Landes de Gascogne is a mixt forest dominated by *Pinus pinaster*. The sampling plot was converted into maize in 1992. The soil is a sandy hydromorphic podzol

(Jolivet et al., 2006) with a clay content of less than 5 % and a sand content higher than 90 %. The first 25 cm are ploughed and crop residues are returned to the soil after each harvest.

Leptosol: the 5-10 cm surface layer of a mollic leptosol was sampled from the long-term Ecosystem Research experiment "Oak Observatory at Observatory of Haute-Provence" (O3HP), France in July 2014. The vegetation is dominated by *Quercus pubescens*. This soil is derived from limestone, compact and iron-rich with clay content

(mainly smectite) of 54%.

### 2.2 Soil incubation

### 2.2.1 Soil preparation






Soil samples were air-dried at 20°C and sieved to 2 mm. Residual soil moisture was determined in parallel by oven-drying an aliquot at 105°C. Thirty-five grams of cambisol and podzol and 30 g of mollic leptosol were transferred into 210 mL incubation jars. Each incubator was then moistened with ulta-pure water ($\delta^2H$ = - 55‰) at 24 g g$^{-1}$ of dry soil for cambisol and leptosol and 10 g g$^{-1}$ for podzol before pre-incubation at 28°C in the dark

for 10 days to re-establish the biological activity to the basal level and to avoid confusion between rewetting-induced and substrate-induced activity.

### 2.2.2 Substrate incubations

Glucose, palmitic acid, phenylalanine and isoleucine were introduced separately in different incubators. They

represent the most common primary compounds of the glucide, lipid and protein families found in either plant or microbial matter and contain different functional groups.

$^{13}C$-labeled and $^2H$-labeled molecules and $^2H_2O$ were provided by Euriso-Top (Cambridge Isotope Laboratories, Inc., Andover, England).

The isotopic abundance of each molecule was adjusted to the desired value by mixing labeled and unlabeled

sources. We prepared "$^{13}C^2H$" (double-labeling) solutions and "$^{13}C^1H$" (mono-labeling) solutions for all molecules. The incubation characteristics are shown in Table S1 (supplementary material). Mixing was performed gravimetrically.

For palmitic acid, the equivalent amount of unlabeled and labeled compound was added to 200 mg of soil and was melted at 70°C. We finely ground the cooled mixture to obtain a homogenized powder that could be added

to the incubators. Two powders were prepared: a $^{13}C^2H$-enriched powder and a $^{13}C$-enriched powder.

Three distinct labeling experiments were performed:

1)   Experiment 1: "$^{13}C^2H$ + H$_2$O": Double-labeling molecule introduced to the soil with ultra-pure water.

2)   Experiment 2: "$^{13}C^1H$ + $^2H_2O$": Mono-labeling molecule introduced to the soil with deuterated water.

3)   Experiment 3: "no molecule + $^2H_2O$": Only deuterated water introduced to the soil.

The final humidity of the soil was 30 g g$^{-1}$, 15 g g$^{-1}$, and 31 g g$^{-1}$ dry weight, respectively, for cambisol, podzol and leptosol.

### 2.2.3 Incubations

The 300 incubators were incubated at 28°C in the dark, and three were frozen at 0, 7, 14, and 28 days and 1 year

for the cambisol and at 0 and 7 days and 1 year for the two other soils. Jars were briefly opened every two days during the first three weeks and then every week until the end of incubation to keep the system under aerobic conditions. Control incubators were prepared for each experiment at each time without any added substrate or deuterated water under the same incubation conditions.

To highlight the link between the NEH and carbon dynamics we initially added three different amounts of

labeled glucose to the podzol and we analyzed the results after 7 days of incubation.

### 2.3  Lipid extraction

Lipids were extracted to isolate the organic non-exchangeable hydrogen. Lipids extractions were performed on



samples that had received glucose and had been incubated for 1 year. Between 10 and 15 g of soil were sub-sampled and phosphate buffer, chloroform and methanol were added (0.8:1:2; v:v:v). After 2 min of ultrasonic and 30 min of warming (37°C), samples were centrifuged for 8 min a 2600 tr min-1. Supernatant was retrieved and stored at room temperature while chloroform and methanol (1:2; v:v) were added to the remaining soil and centrifuged again. Supernatant was retrieved and added to the previous aliquot. Twenty ml of NaCl were then added to the supernatant to distinguish two phases. The denser part was collected and dried under nitrogen.

2.4  Isotopic measurements

Prior to analysis, incubated samples were freeze-dried, ground to a fine powder and kept in closed tubes under laboratory atmosphere. Labeled and unlabeled samples were kept under the same atmosphere until the final $\delta^2H$ measurement. Twenty to fifty-five milligrams of soil were then introduced into a 10 mm tin capsule.

Lipid samples were solubilized in dichloromethane before introducing them to the tin capsules. We let the solvent evaporate before the analysis. The mean isotopic signature of this bulk lipid fraction was measured using the same method as for the soil samples.

The $^{13}C$ and $^2H$ contents were analyzed simultaneously with a combustion module-cavity ring-down spectroscopy (CM-CRDS) isotope analyzer (Picarro, B2221-i). The organic standards polyethylene (IAEA CH7; $\delta^{13}C$ = - 32.15 ± 0.05‰; $\delta^2H$ = - 100.3 ± 2.0‰) and oil NBS-22 ($\delta^{13}C$ = - 30.03 ± 0.05‰, $\delta^2H$ = - 119.6 ± 0.6‰) were used to calibrate the measurements. A homemade standard (olive oil) was also used in each run ($\delta^{13}C$ = - 29.0 ± 0.2‰, $\delta^2H$ = - 153 ± 5 ‰). To validate the measurement on highly enriched samples, we verified the theoretical isotopic composition of the initial conditions of the incubators by calculation.

The isotopic composition of $^{13}C$ and $^2H$ are expressed by abundance (A) or as δ (‰)

$A^{13}C = {^{13}C}/({^{13}C}+{^{12}C})$ and $A^2H = {^2H}/({^2H}+{^1H})$

$δ‰ = [R_{sample} / R_{standard} - 1]*1000$

where R= $^{13}C/^{12}C$ or $^2H/^1H$. The international standard was VPDB for carbon and VSMOW for hydrogen.

2.5  Mass balance equations

Table 2 summarizes the different variables used in the mass balance equations and supporting information (S2) is provided for further understanding of calculations.

The carbon isotopic composition of the total bulk soil corresponds to the proportion of molecule-derived (labeled) and soil-derived (unlabeled) carbon (Eq. (1)).

$$C_{dfm} = ({^{13}A_{tot}} - {^{13}A_{tot\_0}})/({^{13}A_m} - {^{13}A_{tot\_0}})* C_{tot} \qquad (1)$$

Exchangeable hydrogen has the isotopic composition of the atmosphere when the sample is dry. Non-exchangeable hydrogen from the labeled source was estimated based on the simultaneous measurement of labeled and unlabeled samples equilibrated with the same atmosphere using the equations (2) and (3), which attributes all the excess deuterium (difference between the $^2H$ abundances of the labeled sample and unlabeled control) to the NEH derived from the labeled source atoms (see supplementary material S2 for the calculation).

In Experiment 1 (labeled molecule):

$$H_{dfm} = (A_{tot} - A_{tot\_0})/(A_m - A_{tot\_0})*H_{tot} \qquad (2)$$

In Experiment 2 (labeled water):



$$H_{dfw}= (A_{tot} - A_{tot\_0})/(A_w - A_{tot\_0})*H_{tot} \qquad (3)$$

The labeled source is highly enriched compared to natural soil or water: $^{13}A_m$=6.08 to 16.08 %, $A_m$ = 2 to 3.5 % and $A_w$= 0.26 % (supplementary material, Table S1).

The mean $^2H$ abundance of unlabeled soil and water is approximately 0.015 %, and the mean $^{13}C$ abundance is approximately 1.08 %.

Uncertainties in the element and isotope ratio measurements affect the estimate of the amount of labeled-source–

derived carbon or hydrogen atoms. To assess the uncertainty in the calculated values $H_{dfm}$ and $C_{dfm}$, we calculated the statistical error propagation of the uncertainties of the measured isotopic compositions and the element content of the replicated samples (Supplementary material S3).

## 3 Results


### 3.1 Comparison of the four substrates mineralization

The fates of labeled C or H atoms are presented as the mass of C or NEH derived from the labeled source, i.e. molecule or water, $C_{dfm}$, $H_{dfm}$, $H_{dfw}$; eq. (1), (2), (3) (dfm and dfw stand for derived from the molecule and

derived from water, respectively). We first tested the dependence on time (1 week and 1 year), molecule type and soil type on the basis of a three-way ANOVA of each explained variable. Both $C_{dfm}$ and $H_{dfw}$ were dependent on time (p<0.001) and soil (p<0.001) but not on molecule. $H_{dfm}$ was dependent on time (p<0.001), soil type (p<0.001) and molecule (p<0.001). The differences in results for $H_{dfm}$ can be explained by the uncertainty in experiments and measurements. Because we found no significant differences between the molecules for $C_{dfm}$ and

$H_{dfw}$, we considered the different molecule incubations as replicates to simplify the results presentation, with only the mean values shown in graphs. The difference in degradation between molecules is therefore contained in the error bars in fig. 1 and 2.

### 3.2 Carbon mineralization


In fig.1, the results of the carbon and hydrogen derived from the molecule are expressed in percent: the amount of $H_{dfm}$ and $C_{dfm}$ calculated in Eq. (6) and (7) relative to the amount of $H_m$ and $C_m$ in the added molecule. The degradation of the added molecule was very fast. After 7 days, 42 %, 31 % and 53 % of molecule-derived carbon remained in the cambisol, podzol and leptosol, respectively (fig. 1). This trend is in agreement with previous

studies (Murayama, 1988; Derrien et al., 2007) and illustrates the almost complete consumption of the substrate in a few days. Approximately 30 to 50 % of the consumed material was converted into microbial products, and the remaining part was used for heterotrophic respiration. During the following months, the mineralization of organic carbon continued due to the partial consumption of the newly formed microbial carbon by the soil food web.

During the incubation, non-labeled carbon (soil-derived carbon) also decreased by 1.8, 2.4 and 4.5 mg g$^{-1}$ within one year in the cambisol, podzol and leptosol, respectively.

### 3.3 Molecule-derived non-exchangeable hydrogen



After incubation, molecule-derived NEH ($H_{dfm}$) was considerably lower than molecule-derived carbon for the
cambisol and for the podzol (fig. 1): expressed in % of the initial labeled NEH, it was, 12 % and 5 % compared
to the 42 % and 31 % recorded for carbon. It is important to note the different fate of the leptosol, where the
yield of transfer of NEH reached 55 % after 7 days of incubation (fig. 1). During the following months, $H_{dfm}$
slightly decreased in the three soils (by approximately 6 % ± 5 after one year).


3.4  Incorporation of water hydrogen

Experiment 2 (molecule $^{13}C^1H$ + $^2H_2O$) highlights the incorporation of water hydrogen in the non-exchangeable
pool of soil. For the three soils, the incorporation of hydrogen from water tremendously increased during the first
seven days and continued to slowly increase during the incubation year (fig. 2). Respectively, 0.013±0.001,
0.008±0.002 and 0.33±0.07 mg g$^{-1}$ of hydrogen derived from water was found after 7 days of incubation, and
0.06±0.03, 0.023±0.004, and 0.845±0.003 mg g$^{-1}$ was found after one year of incubation for cambisol, podzol
and leptosol (fig. 2).

Figure 3 shows the difference in the incorporation of water hydrogen with and without added substrate. The
incorporation of water-derived hydrogen was higher when associated with substrate addition. It was twice as
high for the podzol after 7 days of incubation. Figure 4 illustrates that this enhancement of incorporation of water
hydrogen was linearly dependent on the amount of the substrate added to the soil.

3.5  Isotopic composition of lipids


Carbon and hydrogen isotopic compositions of bulk lipids at 365 days for the control soil are presented in Table
3. The proportions of labeled carbon and hydrogen were calculated as the proportion of the total lipid carbon and
hydrogen (($C_{dfm}$/C)$_{lipids}$, ($H_{dfm}$/H)$_{lipids}$, and ($H_{dfw}$/H)$_{lipids}$) and were compared to the proportion in the bulk soil
(($C_{dfm}$/C)$_{bulk}$, ($H_{dfm}$/H)$_{bulk}$, ($H_{dfw}$/H)$_{bulk}$). The $\delta^{13}C$ and $\delta^2H$ of the lipids in the control samples were lower than that
of the bulk soil, in agreement with previous work, where the lipid $\delta^{13}C$ was 2-3‰ lower than the bulk $\delta^{13}C$
(Chikaraishi and Naraoka, 2001; Hayes, 2001), and the lipid $\delta^2H$ was 150‰ lower than the bulk $\delta^2H$ (Sessions et
al., 1999; Chikaraishi and Naraoka, 2001). The average measured H/C ratio of the lipids of the three soils was
2.1 (molar ratio).

The proportion of molecule-derived carbon in the lipids was 1.0, 0.4 and 0.8 % for the cambisol, podzol and
leptosol, respectively, compared to the corresponding values of 0.6, 0.4, and 0.4 % in the bulk organic carbon.
The proportion of molecule-derived hydrogen was, respectively 0.10, 0.02 and 0.19 % for the cambisol, podzol
and leptosol (Table 3). These values were on the same order of magnitude as the molecule-derived hydrogen in
the bulk soil. The proportion of labeled water-derived hydrogen was 1.0, 0.4 and 1.1 % of the total hydrogen
content in the lipids and 1.5, 0.9 and 6.8 % in the total bulk soil for the cambisol, podzol and leptosol,
respectively.

4    Discussion

4.1 Preservation of the organic substrate hydrogen in biosyntheses




The microbial activity is initiated during the first days after the addition of the substrate. The added molecule regardless of its quality is quickly metabolized (fig. 1).

We independently traced the preservation of organic hydrogen (experiment 1) and the incorporation of water-derived hydrogen (experiment 2) during decomposition and biosynthesis. The conservation of organic hydrogen

from the initial substrate is very low in both the total and lipid NEH. The carbon-hydrogen bonds are broken during decomposition, and exchange with exchangeable hydrogen can occur. The difference between carbon and hydrogen isotopic fates during the first seven days (fig. 1) reflects the exchange of hydrogen with water during the early stage of degradation. Subsequently, new organic exchangeable hydrogen derived from water can be incorporated into the non-exchangeable pool of organic matter by biological processes. Furthermore, because

mineralization of a substrate also results in $^2H_2O$ release, one part of the soil organic non-exchangeable $^2H$ may originate from incorporation of this substrate-derived deuterated water into the non-exchangeable pool. Using the assumption that water is a well-mixed isotopic compartment, this amount is between 3 and 5 % of the residual hydrogen from the organic substrate at 365 days. The isotopic composition of the non-exchangeable organic hydrogen is mainly determined by the water isotopic composition (fig. 2). It is in accordance with the work of

Baillif and colleagues that have grown fungi with labeled glucose, water or acetate to trace the incorporation of $^2H$ during fatty acid biosynthesis. They have demonstrated that water is the main donor of hydrogen atoms in the non-exchangeable pool within the biosynthesis cycle (Baillif et al., 2009). Ruppenthal and colleagues have shown as well that precipitation contributes to 80 % of the isotopic composition of non-exchangeable hydrogen (Ruppenthal et al., 2010). Moreover, the incorporation of water hydrogen is favored by the strength of the C-H

bond breakage. It can be weak or strong, depending on the enzyme activated for the degradation of the molecule and the position of the bond (Augusti et al., 2006). When C-H breakdown is favored, the surrounding water imprints its hydrogen isotopic signature on the former bounded-H (Augusti et al., 2006). In the present experiment, we show that more than 70 % of the H-C bonds are broken; therefore, the added molecules are used more for energetic and trophic resources than as building blocks in the biosynthesis.

Deuterium can also accumulate in hydration shells, which have stronger hydrogen bridges than the biomolecule (Baumgärtner and Donhaerl, 2004). The accumulation of deuterium from water occurs in the biomatter during biological processes but also during the hydration of molecules (Baumgärtner and Donhaerl, 2004; Turner et al., 2009).

4.2  Carbon-driven acquisition of the non-exchangeable hydrogen isotope signature

The rapid mineralization of hydrogen from the molecule is due to biodegradation whereas the rapid incorporation of water during the first seven days of incubation is associated with biosynthesis (figs. 1 and 2). Carbon mineralization fosters the formation of non-exchangeable hydrogen from water (figs. 3 and 4). There is also an

incorporation of hydrogen from water in the soil in the experiments without substrate. This could be due to the mineralization of carbon already present in soil (fig. 3). However, in fig. 2, exchange of hydrogen with water seems to be continuous: the incorporation of water hydrogen into the non-exchangeable pool continues increasing, following carbon mineralization.

Results of lipids isotopic compositions show that the amount of newly formed NEH (% of $H_{dfw}$ + % of $H_{dfm}$ in





lipids; Table 3) is slightly higher than expected from the theoretical organic C-H bond (% of $C_{dfm}$ in lipids; Table 3). This could be due to complete, stoichiometric labeling of newly biosynthesized lipids, i.e., lipids formed on the labeled organic carbon plus a smaller amount of newly synthesized lipids from unlabeled organic matter. The proportion of molecule-derived carbon is higher in the lipid than in the bulk soil, and the lipids are derived mainly from microbial biosynthesis. However, the proportions of $H_{dfm}$ in the lipids and in the bulk soil are of the

same magnitude for the three soils (Table 3): hydrogen is derived evenly from the labeled molecule and from the unlabeled soil during lipids biosynthesis. The proportion of hydrogen derived from the water in lipids is lower than the respective proportion in the bulk soil, which means the proportion of NEH derived from water is not necessarily organic. The difference is even higher in the clayey soil (leptosol, Table 3).

In lipids, hydrogen corresponds to the organic, non-exchangeable hydrogen. The hydrogen in lipids formed from

water ($H_{dfw}$) is only organic, whereas in the bulk soil, the non-exchangeable hydrogen is organic and inorganic. To estimate the proportion of organic non-exchangeable hydrogen, we assume that $H_{dfw}/C_{dfm}$ in lipids (0.20 on average for the three soils based on mass ratio, Table 3) is approximately the same as the $H_{dfw}/C_{dfm}$ in the organic fraction of the bulk soil. Using the measured $C_{dfm}$ in bulk soil, we can then estimate the total organic $H_{dfw}$ as 0.014, 0.015 and 0.028 mg $g^{-1}$, respectively, for the cambisol, podzol and leptosol. The proportion of inorganic

NEH is therefore 0.046, 0.008 and 0.82 mg $g^{-1}$ for the three soils. The NEH isotopic composition is mainly controlled by the incorporation of water through biosynthesis, but the inorganic NEH is not negligible, especially in the clayey soil.

4.3  Hydrogen dynamics in different soil types.


The association of organic matter with minerals is known to decrease the decomposition rate of the former (Feng et al., 2013; Jenkinson and Coleman, 2008; Vogel et al., 2014). This result is observed in our experiment by comparing the three soils with increasing clay content and is applicable to both H and C in both bulk soil (fig. 1) and lipids (Table 3). However, the clay content has an important role in the incorporation of water-derived

hydrogen beyond this organic matter stabilization effect. In clayey leptosol, the amount of labeled NEH from the molecule ($H_{dfm}$; fig. 1) is much higher than in the other soils, which may be explained by the preferential use of hydrogen locally near biological reactions. Hydrogen derived from the mineralization of the substrate does not directly exchange with the total pool of water but with a smaller pool. The resulting local water pool has a less negative isotopic signature than the remaining water pool. Water incorporation through biosynthesis could then

occur with this smaller pool of $^2$H-enriched water. Moreover, hydrogen exchange within the whole water pool is slowed by the presence of clay, which accumulates molecule-derived hydrogen in hydroxyl sites. For this reason, water-derived NEH is also much higher than in the other soils. The NEH pool in leptosol is bigger (confirmed by the calculation of the total organic NEH) due to the inorganic NEH present in the clay fraction. The inorganic NEH may exchange very slowly on a longer timescale, but on a short-term dynamics scale, this pool acts as non-

exchangeable and is mostly at the hydroxyl position (López-Galindo et al., 2008).

The zonal distribution of organic compounds associated with minerals (Kleber et al., 2007; Vogel et al., 2014) may control the exchange between soil solution and organic compounds at kinetics that differ according to the layer within the organo-mineral interaction zone. The non-exchangeable hydrogen dynamics in soil organic matter are not independent of the mineral structure. The type of clay plays a role in carbon sequestration,





depending on the specific surface area of the mineral or aggregate (Vogel et al., 2014; López-Galindo et al., 2008). In leptosol, clays are mainly smectite and have a high specific surface area. The high content of iron and hydroxide present in the leptosol also increases the specific surface area of the aggregates, which increases the organo-mineral association (Baldock and Skjemstad, 2000). Organic carbon cycling itself may be associated with mineral transformation (Basile-Doelsch et al., 2015), which may involve the newly formed hydroxyl.

The short-term dynamics of hydrogen are driven by the incorporation of hydrogen from water by isotopic exchange and by microbial biosynthesis. However, the increase in the incorporation of water hydrogen with the soil clay content suggests that part of the hydrogen is bound to clay or organo-mineral complexes. The production of NEH from water occurs mainly during the first weeks, but slow exchange of water hydrogen continues during the following year. Lopez-Galindo *et al.* observed the same trend, and they related the

accumulation rate to the clay mineral properties (López-Galindo et al., 2008).

### 4.4 Ecosystem-scale production and fate of non-exchangeable hydrogen

In the present experiment, the preservation of non-exchangeable hydrogen from an organic substrate is less than

5 % after one year in soil with a low clay content. Water is the main donor of hydrogen during biosynthesis cycle favored by the breakage of the C-H bonds of the initial substrate. In this work, we showed that 70 % of the C-H bonds of the initial substrate were broken during biosynthesis. Concerning the fate of tritium in terrestrial ecosystems, the isotopic composition of the organic plant material is a minor determinant of the bulk soil organic matter composition. However, the preservation of hydrogen from vegetation could increase with the soil clay

content by organo-mineral and zonal interactions. Water will be the main donor of organically bound tritium in the soil dependent to the carbon mineralization. The lipids isotopic composition in this experiment have highlighted that the newly formed non-exchangeable hydrogen is not necessary organic. Therefore, the incorporation of tritium from water in NEH pool is dependent on the clay content in addition to the interaction with the soil hydrodynamics.

In our work, the isotopic composition of the NEH pool is determined by comparing labeled samples with unlabeled samples equilibrated under the same atmosphere. This method includes inorganic NEH. A proportion of the inorganic, non-exchangeable hydrogen should be taken into account in the prediction of the dynamics of hydrogen and tritium.

The proportion of NEH associated with minerals is itself partially related to the carbon dynamics. The long-term

fate of tritium in terrestrial environments will depend on the status of soil carbon dynamics at the moment of tritium contamination.

Both the carbon dynamics and the incorporation of inorganic hydrogen in soils should therefore be taken into account in a conceptual model for the prediction of the long-term fate of hydrogen, and thereafter of tritium, in soil organic matter. The results of the present study can be used for the parameterization of the carbon-hydrogen

coupling in such prediction models.

Acknowledgements: This study was funded by EDF and Andra. We would like to thank Sabine Maestri for her great help with the isotopic measurements. We thank the following research teams for having gently put at our disposition their long-term experiments and associated data: INRA-Agro-Paristech ECOSYS and INRA-UVSQ





GCVG (Les Closeaux SolFit experiment in Versailles), INRA Unité Experimentale Foret Pierroton and INRA
ISPA (Pierroton forest reserve and fertilization experiment); OSU-Pytheas (O3HP LTER in Saint-Michel
l'Observatoire).

Supplementary material includes the summary of incubation characteristics (Table S1), mass balance calculation
(S2) and propagation error calculation (S3).

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



## Tables and Figures

Table 1: Pedologic, geographic information and carbon and hydrogen isotopic composition of the three bulk soils.

| Soil | Latitude, longitude | Sampling depth (cm) | plant cover | % clay | % silt | % sand | C mg g$^{-1}$ | H mg g$^{-1}$ | N mg g$^{-1}$ | pH | Mean bulk δ$^{13}$C (‰) | Mean bulk δ$^2$H (‰) |
|------|---------------------|---------------------|-------------|--------|--------|--------|---------------|---------------|---------------|-----|-------------------------|----------------------|
| Cambisol | 48° 48' 0.2 N; 2° 06' 33 W | 0-25 | wheat | 17 | 50 | 33 | 12 | 3.2 | 1.27 | 6.8 | -26 ± 0 | -53±3 |
| Podzol | 44° 44' 33 N; 0° 47' 37 W | 0-25 | maize | 5 | 3 | 92 | 20 | 2.3 | 0.84 | 5.5 | -26 ± 0 | -51±3 |
| Leptosol | 43° 56' 06 N; 5° 42' 06 W | 5-10 | oak forest | 54 | 37 | 9 | 39 | 13.7 | 3.59 | 7.5 | -25 ± 0 | -52±2 |





Table 2: definition of the variable used in calculations

| | Variables | Definition |
|---|---|---|
| **Quantity** (mg g$^{-1}$ of dry soil) | $H_{tot}$ | Total amount of hydrogen in the soil |
| | $C_{tot}$ | Total amount of carbon in the soil |
| | $H_{dfw}$ | Amount of non-exchangeable hydrogen derived from water |
| | $H_{dfm}$ | Amount of non-exchangeable hydrogen derived from molecule |
| | $C_{dfm}$ | Amount of carbon derived from molecule |
| | $H_m$ | Initial amount of non-exchangeable hydrogen in the added molecule |
| | $C_m$ | Initial amount of carbon in the added molecule |
| **Abundance** | $^{13}A_{tot\_0}$ | $^{13}C$ abundance of the unlabeled experiment (control) |
| | $A_{tot\_0}$ | $^{2}H$ abundance of the unlabeled experiment (control) |
| | $^{13}A_{tot}$ | $^{13}C$ abundance of the total bulk soil |
| | $A_{tot}$ | $^{2}H$ abundance of the total bulk soil |
| | $^{13}A_m$ | Initial $^{13}C$ abundance of the labeled molecule |
| | $A_m$ | Initial $^{2}H$ abundance of the labeled molecule |
| | $A_w$ | Initial $^{2}H$ abundance of the labeled water |






Table 3: $\delta^{13}$C, $\delta^2$H of bulk soil and lipids at 365 days of incubation for cambisol, podzol and leptosol and the proportion of carbon and hydrogen derived from the labeled-source. In brackets the concentration in mg g$^{-1}$ of carbon and hydrogen derived from the labeled-source

| | $\delta^{13}$C ‰ | | $\delta^2$H ‰ | | % and concentration (mg g$^{-1}$) of the labelled source in the bulk soil | | | % and concentration (mg g$^{-1}$) of the labelled source in lipids | | |
| | Bulk | lipids | bulk | lipids | % Cdfm (mg g$^{-1}$) | % Hdfm (mg g$^{-1}$) | % Hdfw (mg g$^{-1}$) | % Cdfm (mg g$^{-1}$) | % Hdfm (mg g$^{-1}$) | % Hdfw (mg g$^{-1}$) |
|---|---|---|---|---|---|---|---|---|---|---|
| | | | | | **Cambisol** | | | | | |
| Experiment 1 | 5.3 ± 0.9 | 19 ± 3 | 179 ± 2 | 133 ± 43 | 0.6 % (0.054 ± 0.009) | 0.082 % (0.0031 ± 0.0003) | - | 1 % (3.3 ± 0.6) | 0.1 % (0.13 ± 0.01) | - |
| Experiment 2 | 5 ± 1 | 17 ± 11 | 422 ± 15 | 36 ± 6 | 0.6 % (0.055 ± 0.009) | - | 1.5 % (0.06 ± 0.03) | 1 % (4.2 ± 0.8) | - | 1 % (0.68 ± 0.09) |
| | | | | | **Podzol** | | | | | |
| Experiment 1 | 13.2 ± 0.7 | 7 ± 5 | 94 ± 39 | -106 ± 42 | 0.4 % (0.07 ± 0.01) | 0.07 % (0.0017 ± 0.0007) | - | 0.4 % (1.8 ± 0.9) | 0.02 % (0.05 ± 0.02) | - |
| Experiment 2 | 7.0 ± 0.7 | 6 ± 7 | 259 ± 25 | -38 ± 55 | 0.4 % (0.07 ± 0.01) | - | 0.9 % (0.023 ± 0.004) | 0.4 % (1.4 ± 0.1) | - | 0.4 % (0.4 ± 0.2) |
| | | | | | **Leptosol** | | | | | |
| Experiment 1 | 22 ± 3 | 97 ± 2 | 723 ± 10 | 458 ± 65 | 0.4 % (0.15 ± 0.03) | 0.24 % (0.029 ± 0.003) | - | 0.8 % (5.9 ± 0.2) | 0.19 % (0.18±0.01) | - |
| Experiment 2 | 25 ± 1 | 76 ± 20 | 798 ± 58 | 23 ± 5 | 0.4 % (0.14 ± 0.03) | - | 6.8 % (0.8 ± 0.1) | 0.8 % (4.8 ± 0.8) | - | 1.1 % (0.95 ± 0.03) |






List of figures:

Figure1: Percentage of non-exchangeable hydrogen and carbon derived from the added molecule during one year of incubation for cambisol, podzol and mollic leptosol. The grey part corresponds to the results from 0 to 28 days. The line corresponds to the mean value calculated at each time for all molecules experiments.

Figure 2: Concentration of non-exchangeable hydrogen derived from the water $H_{dfw}$ and derived from the
molecule $H_{dfm}$ for cambisol, podzol and mollic leptosol from 0 to 365 days. The line corresponds to the mean value calculated at each time for all molecules experiments.

Figure 3: Concentration of non-exchangeable hydrogen derived from water with and without addition of substrate for the cambisol, podzol and leptosol from 0 to 28 days of incubation.


Figure 4: Amount of substrate-derived H and C at 7 days of incubation versus the concentration of substrate as carbon (0.14, 0.29 and 0.43 mg g$^{-1}$). The concentration C in experiment 1 is equal to 0.43 mg g$^{-1}$.





Fig. 1

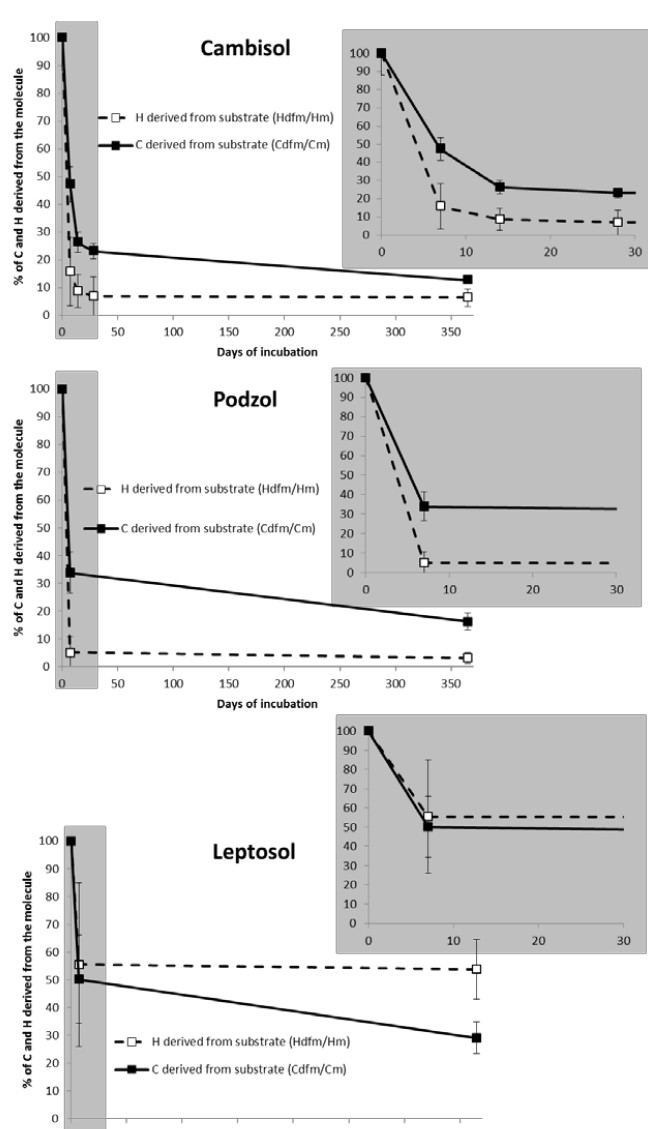






Fig. 2

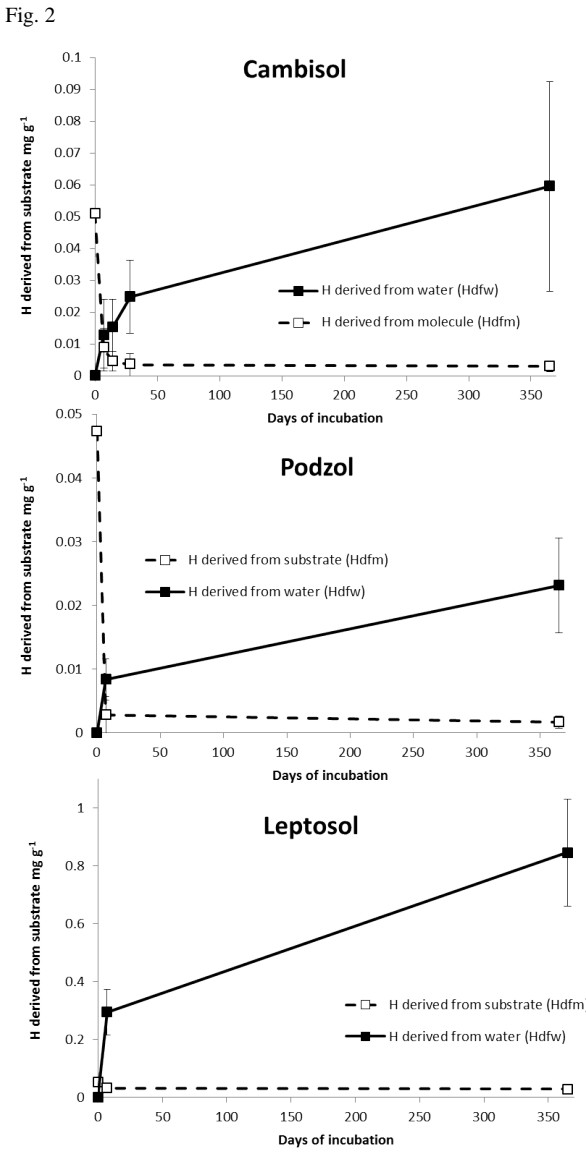





Fig.3

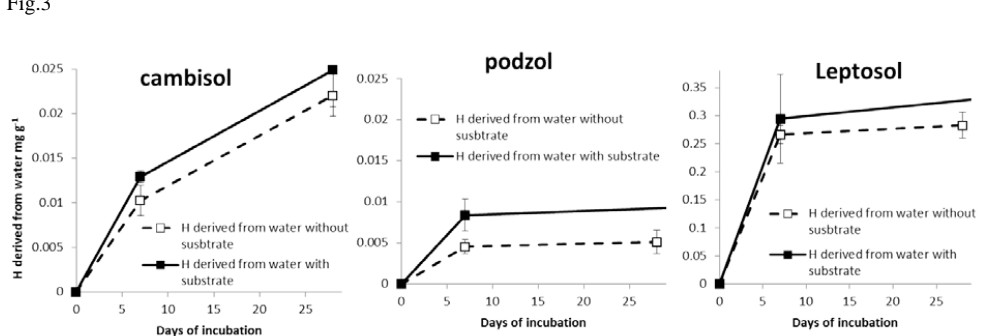




Fig.4

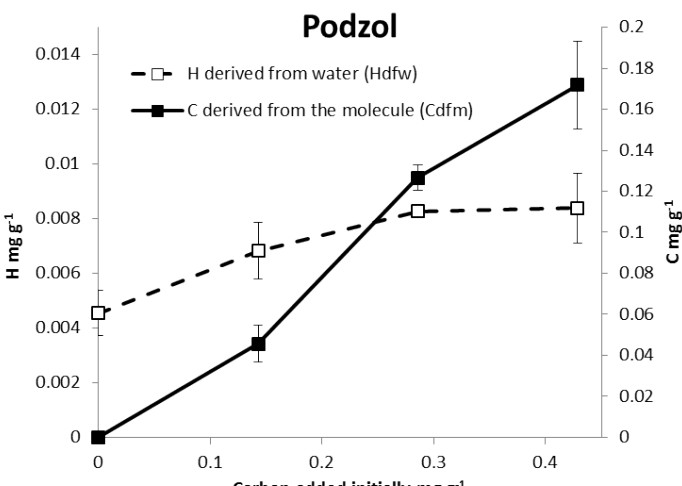