# Peer review of "Hydrogen dynamics in soil organic matter as determined by 13C and 2H labeling experiments"

_Biogeosciences, 2016_

## Referee Comment (RC1) · Anonymous Referee #1 · 4 Oct 2016

This study investigates the hydrogen dynamics in soil organic matter to quantify processes such as the preservation of organic matter and microbial biosynthesis. Seems to be that this research is potentially useful to understand the fate of tritium ($^3$H) in ecosystems. The approach described in this paper for determining the fate of hydrogen in soil systems using three types of labelling experiments (substance, substance/water, only water) is an original approach. However, my major comment is related to the assumptions used for hydrogen exchangeability, which were poorly explained. I believe this manuscript needs significant explanation about the hydrogen isotope analyses and modelling. I therefore recommend publication only after major revisions.

H exchangeability – Soil organic matter could be a heterogeneous material in terms of hydrogen exchangeability. Uncontrolled isotopic exchange between sample and laboratory ambient vapour can introduce bias in $\delta^2$H measurements. The authors did not explicitly account for H exchangeability in their analysis by using the Comparative Equilibration method or the aid of devices that allows vapour equilibration before analysis. Moreover, bulk soil samples without lipid extraction was conducted. As the authors pointed out, lipids do not usually exchange with atmospheric vapour because of the C-H bonds in their main structure. However, differential lipid content in bulk soil might bias the $\delta^2$H measurements as well. In this study, non-exchangeable standards of non-similar matrix to the samples were run for calibration and hydrogen exchangeability seem to be corrected by measuring labeled and unlabeled samples at the same time. In theory, this could be a reasonable way to deal with this issue, but the authors should provide more details.

Specific comments

p4, line 106: Residual soil moisture is of great relevance when determining H isotope measurements because it would be a reservoir of H in the sample to be analyzed. Was it estimated once at the beginning of the experiment? Was performed after collection from incubators and freeze-drying? This step is crucial to eliminate any 'contamination' of residual moisture from the experiments.

p4, line 109: Please confirm amounts of water added.

p4, line 108: Provide uncertainty associated with this value.

p4, lines 134-140: One striking thing is the incubation experiment protocols. The authors opened the incubation systems every two days during the first three weeks and then every week. I understand this is important to keep aerobic conditions along the experiment. Would this compromise the $^2$H abundance of the water? Further explanation is required here.

p5, line 147: For how long were samples freeze-dried? Again, this step is further relevant to eliminate possible contamination of 'deuterated' moisture in the sample to be analyzed. Previous investigations with organic materials have found that long periods of drying are needed.

p5, lines 153-162: Needs a more detailed description of the analyses. For example, a merit of precision using this method based on the standards measured is needed. How the $^2$H abundance of water was measured?

More importantly, how the authors deal with the hydrogen exchangeability is quite reduced in the manuscript and relatively obscured to the reader. In the section 2.4, the authors only stated the following sentence: "Labeled and unlabeled samples were kept under the same atmosphere until the final $\delta^2$H measurement." Would that mean that they conducted a comparative equilibration method? This method is extensively used in the literature, but mostly for natural abundance samples. Any modifications for labeled samples are required? How long were the samples left under the same atmosphere? Which atmosphere? Laboratory atmosphere? Or inside a desiccator and then opened to the laboratory ambient? In short, the authors need to provide more details in their methodological section.

Another question I have is whether the use of two reference materials for calibration that cover a very small range of delta values (~2 per mil for $\delta^{13}$C and ~20 for $\delta^2$H) can adversely affect the accuracy of their measurements of labeled samples among runs. Do the authors consider the use of a labelled standard?

Were other analytical issues such as memory effects considered? Previous published studies (i.e. Koehler and Wassenaar 2012 Anal Chem) that describe this type of technique for natural abundance samples (combustion + laser spectroscopy) have shown some measurable memory issues for hydrogen, at least. I suspect that labeled samples should be further affected.

p8, lines 271-273: How was this amount calculated?

p8, lines 282-283: "In the present experiment, we show that more than 70 % of the H-C bonds are broken". Is this correct in view of the H exchangeability concern during analysis?

Figures. During the whole manuscript I missed the results of $^{13}$C and $^2$H abundances of the bulk soil and lipids during the length of the labeled and unlabeled experiment. Specially, when the correction of hydrogen exchangeability seem to be performed by measuring the labeled and unlabeled samples.

Figure 2. In the hydrogen labeling experiments performed, there are two sources of hydrogen: substrate and water. In relation to the mineralization of labeled substrates is clear to me since a starting amount of molecule (day 0) became consumed along the experiment and the labeling signature is decreased. However, for the water, it is a different story. I believe the incubators used were filled with plenty of labeled water, which means the labeled signature never is consumed or decreased. I wonder if the trend of estimated H derived from water in this figure is based on the isotopic equilibrium with the labeled water instead of an observed derivation of H from water in vivo into microbial biosynthesis. Again, as previous comment, showing the measured $^2$H abundances over the length of experimentation could provide insights to clarify these points.

Table 3. One more noticeable thing in the table is that the results between H (% Hdfw) and C (% Cdfm) are quite consistent in lipids, which do not have exchangeable hydrogen. In the other hand, the proportion of hydrogen derived from the labeled source did not behave similarly in the bulk soil (with exchangeable H). A clear explanation on the treatment of exchangeable hydrogen can convince the reader on a differential isotopic routing of H and C.

---

## Referee Comment (RC2) · Anonymous Referee #2 · 11 Oct 2016

This is an important paper in the context of understanding of hydrogen dynamics in soil organic matter. The paper is well written and sound. I believe the backline story about Tritium detracts from the main study research outcomes as the context of tritium remains only touched upon, as we get no real concrete data about the concerns raised line 64/65. There is mention of enhancement tritium entering the environment due to historic bomb-testing but no mention that tritium is a radioactive form of H (half-life 12.3 years), unlike 2H and 1H who are stable isotopes, so will over time will dissipate and has done already decline since the bomb-14C peak. The authors should either reduce the tritium context or make it more quantitative. The authors may wish to comment on the potential of water in the air (different isotopic H signature) to enter the experimental jars and when opening them to prevent anaerobic conditions occurring in the jar tremendously (line 229) maybe find another word to describe the vary large increase

observed. For Figure 2 and 3 the scale on the y-axis between the various soil types are different, which makes immediate comparisons difficult. If, the authors want to retain this, maybe indicate in the legends of Figure 2 and 3 that this is the case "note, the scale of Y-axis varies between the subfigures'.

---

## Author Comment (AC1) · 3 Nov 2016

We thank the referee for the relevant and constructive comments and we appreciated the quality of the observations, which allowed us to improve the first version of our paper.

The referee comments are reported below with the answers we provided.

**Anonymous referee#1**

*This study investigates the hydrogen dynamics in soil organic matter to quantify processes such as the preservation of organic matter and microbial biosynthesis. Seems to be that this research is potentially useful to understand the fate of tritium (3H) in ecosystems. The approach described in this paper for determining the fate of hydrogen in soil systems using three types of labelling experiments (substance, substance/water, only water) is an original approach. However, my major comment is related to the assumptions used for hydrogen exchangeability, which were poorly explained. I believe this manuscript needs significant explanation about the hydrogen isotope analyses and modelling. I therefore recommend publication only after major revisions.*

1. *H exchangeability – Soil organic matter could be a heterogeneous material in terms of hydrogen exchangeability. Uncontrolled isotopic exchange between sample and laboratory ambient vapour can introduce bias in δ2H measurements. The authors did not explicitly account for H exchangeability in their analysis by using the Comparative Equilibration method or the aid of devices that allows vapour equilibration before analysis.*

   Answer: We agree with the referee that some details of the methodology are missing in this work, especially to deal with the exchange of hydrogen. In this study, we did not use devices for equilibration but we compared the isotopic composition of unlabeled and highly labeled samples of soils after equilibration of the dried samples with laboratory atmosphere. No standards were chosen in the comparison but we assumed that unlabeled and labeled samples exposed to the same atmosphere before $\delta^2H$ measurement reach the same concentration in deuterium in the labile hydrogen pool. Then calculating the difference in $^2H$ abundance between unlabeled and labeled samples allows eliminating the contribution of labile hydrogen (short-time exchange) in the final isotopic calculation. This difference represents the isotopic composition of hydrogen that did not exchange during the length of equilibration. It is called non-exchangeable hydrogen in this paper. See also comments 11.

2. *Moreover, bulk soil samples without lipid extraction was conducted. As the authors pointed out, lipids do not usually exchange with atmospheric vapour because of the C-H bonds in their main structure. However, differential lipid content in bulk soil might bias the δ2H measurements as well.*

   Answer: In our study, lipids are extracted to provide, through their excess $^{13}C$/excess $^2H$, a proxy of the ratio of organic HNE biosynthesis/ C biosynthesis. In these soils, extractable lipids (by conventional CHCl3-MeOH method) accounts for ca 1.5 % of

soil C, whereas total C-H (sum of aliphatic, sugar C-H and aromatic C account for more than 50% of soil organic C (as determined from CP-MAS 13C NMR of one of the soils). We therefore consider that differences in lipids content between soils would not affect the results more than differences in bulk organic carbon content. Moreover, the isotopic results of the labeled lipids are always corrected from the extracted unlabeled lipids to interpret only the excess $^{13}$C/ excess$^{2}$H.

3. *In this study, non-exchangeable standards of non-similar matrix to the samples were run for calibration and hydrogen exchangeability seem to be corrected by measuring labeled and unlabeled samples at the same time. In theory, this could be a reasonable way to deal with this issue, but the authors should provide more details.*

> Answer: We agree and improved it: See comments 11

*Specific comments*

4. *p4, line 106: Residual soil moisture is of great relevance when determining H isotope measurements because it would be a reservoir of H in the sample to be analyzed. Was it estimated once at the beginning of the experiment?*

> Answer: The residual soil moisture was estimated once before adding water and substrate.

> Action: The residual soil moisture is added to the manuscript section 2.2.1. p.4

5. *Was performed after collection from incubators and freeze-drying? This step is crucial to eliminate any 'contamination' of residual moisture from the experiments.*

> Answer: We determined the end of freeze-drying when the weight of the sample reached its initial dry weight. See also comments 8.

> Action: Dry residual soil moisture were added in the section 2.2.1, p. 4

6. *p4, line 109: Please confirm amounts of water added.*

> Action: We completed in line 112/113 p. 4 the amount of added water to reach the humidity required for the incubation.

7. *p4, line 108: Provide uncertainty associated with this value.*

> Action: The uncertainty of the isotopic composition is added line 111 p. 4 ($\delta^{2}$H = -63.8 ± 0.5 ‰).

8. *p4, lines 134-140: One striking thing is the incubation experiment protocols. The authors opened the incubation systems every two days during the first three weeks and then every week. I understand this is important to keep aerobic conditions along the experiment. Would this compromise the $_2$H abundance of the water? Further explanation is required here.*

> Answer: By taking the value of the saturation vapor pressure at 28°C (28 g/m$^3$), the amount of water contained in the headspace jar (0.17 dm$^3$) was 4.8 mg. The proportion of labeled water vapor lost by the renewal of incubation jar headspace was estimated at 0.7 % the first months and 2 % at one year. The impact of the atmosphere renewal on the isotopic composition was therefore neglected.

> Actions: We provided information in section 2.2.3, p. 4/5

9. *p5, line 147: For how long were samples freeze-dried? Again, this step is further relevant to eliminate possible contamination of 'deuterated' moisture in the sample to be analyzed. Previous investigations with organic materials have found that long periods of drying are needed.*

> Answer: We agree this is an important question. The efficiency of freeze drying is highly dependent on sample volume and geometry, gas flows, gas pressures, and sample temperature. We optimized these four factors. Samples were freeze-dried for 28 hours. We conducted tests that revealed a constant weight after 16 hours in our conditions. Final sample temperature was 24°C and final vapor pressure < 0.1 mbar.

> Action: We completed the text line 156 p.5

10. *p5, lines 153-162: Needs a more detailed description of the analyses. For example, a merit of precision using this method based on the standards measured is needed. How the $_2$H abundance of water was measured?*

> Answer: The $^2$H abundance of labeled water was calculated not measured. The isotopic composition of the deuterated solution added in the soil was calculated and adjusted during the dilution step.

11. *More importantly, how the authors deal with the hydrogen exchangeability is quite reduced in the manuscript and relatively obscured to the reader. In the section 2.4, the authors only stated the following sentence: "Labeled and unlabeled samples were kept under the same atmosphere until the final $\delta_2$H measurement." Would that mean that they conducted a comparative equilibration method? This method is extensively used in the literature, but mostly for natural abundance samples. Any modifications for labeled samples are required? How long were the samples left under the same atmosphere? Which atmosphere? Laboratory atmosphere? Or inside a desiccator and then opened to the laboratory*

*ambient? In short, the authors need to provide more details in their methodological section.*

Answer: We totally agree with this comment. We were unclear and the equilibration is in itself the definition of NHE. Together with hydrogen bound to carbon, non-exchangeable hydrogen may include other species with exchange rate depending on organo-mineral and mineral dynamics.

The definition of NEH depends on the method used for equilibration, from simple atmospheric equilibration (Wassenaar and Hobson, 2003) to high pressure and high temperature equilibration with water vapor (Schimmelmann, 1991). The definition of NHE in this study corresponds therefore to hydrogen in dry soil that didn't exchange with atmosphere during the equilibration phase.

We equilibrated unlabeled and labeled samples with the lab atmosphere for 2 hours after soil grinding (exchanges also occur during the grinding $\approx$ 20 min and during the evaporation by nitrogen flushing in the CM-CRDS introduction line).

The differences of $\delta^2H$ between unlabeled and labeled samples are a mean to eliminate the contribution of labile hydrogen (short-time exchange) in the final isotopic calculation. Unlabeled and labeled sample received exactly the same treatment. When compared to Comparative Equilibration method , the absolute $\delta^2H$ of NHE is not quantified, but it is calculated and is equal to the relative enrichment of labeled vs. unlabeled sample is similar (see mass balance calculation section 2.5.), because both are equilibrated with the same atmosphere. One advantage of our method is that no standard material is needed for NHE quantification.

Considering your question of adaptation to labeled samples, there is no theoretical consideration that would differentiate highly labeled samples. Considering the sensitivity of the method, in our case, the difference in $\delta^2H$ between the equilibration source (natural) and sample (labeled) is very high, so that the sensitivity would be better than using natural samples.

Action: We added a specific section to the attention of the reader in section 2.5.1 p. 6 to explain further the equilibration method we used and the definition of non-exchangeable hydrogen we consider.

12. *Another question I have is whether the use of two reference materials for calibration that cover a very small range of delta values (~2 per mil for $\delta_{13}C$ and ~20 for $\delta_2H$) can adversely affect the accuracy of their measurements of labeled samples among runs. Do the authors consider the use of a labelled standard?*

Answer: We did not use labeled standard. However, to validate the measurement of highly enriched samples, and the linearity, we calculated the theoretical abundance of the labeled samples at the initial condition, before degradation. Measured values are compared to the theoretical values using the $^{13}C$- $^2H$- labeled organic substrates (slope of 1.02 and r² = 0.99).

Action: We added information in section 2.4. p.5

13. *Were other analytical issues such as memory effects considered? Previous published studies (i.e. Koehler and Wassenaar 2012 Anal Chem) that describe this type of technique for natural abundance samples (combustion + laser spectroscopy) have shown some measurable memory issues for hydrogen, at least. I suspect that labeled samples should be further affected.*

Answer: To deal with the $^2$H memory effect often recorded with CM-CRDS, 5 repetitions were done for each sample. The last three were used for interpretation when standard deviation was less than 3 ‰ for natural samples and less than 10 ‰ for enriched samples. Moreover, we analyzed samples from the more depleted to the more enriched and ashes samples were removed from the combustion tube each 45 samples (maximum) to limit contamination.

Action: We added explanations line 174-178 p. 5/6.

14. *p8, lines 271-273: How was this amount calculated?*

Action: We added the calculation in supplementary material. Calculation is shown at the end of this document as well.

15. *p8, lines 282-283: "In the present experiment, we show that more than 70 % of the H-C bonds are broken". Is this correct in view of the H exchangeability concern during analysis?*

Answer: The conservation of carbon from the molecule is higher than the conservation of non-exchangeable hydrogen from the molecule during the length of incubation suggesting that the initial C-H bonds of the molecule are broken (fig. 1). The exchange of hydrogen is then possible.

Action: We added details lines 336-337 p.9

16. *Figures. During the whole manuscript I missed the results of $_{13}$C and $_2$H abundances of the bulk soil and lipids during the length of the labeled and unlabeled experiment. Specially, when the correction of hydrogen exchangeability seem to be performed by measuring the labeled and unlabeled samples.*

Answer: We agree that such data are missing. Because these would require 80 kinetic curves, that bring no more information, we put these curves as supplementary material. The curves are also shown at the end of this document.

Action: In the text, we indicated the magnitude of $\delta^{13}$C and $\delta^2$H signatures, for the reader to catch the high difference between labeled and natural samples at the beginning of the Results section: Lines 226-241 p. 7.

17. *Figure 2. In the hydrogen labeling experiments performed, there are two sources of hydrogen: substrate and water. In relation to the mineralization of labeled substrates is clear to me since a starting amount of molecule (day 0) became consumed along the experiment and the labeling signature is decreased. However, for the water, it is a different story. I believe the incubators used were filled with plenty of labeled water, which means the labeled signature never is consumed or decreased. I wonder if the trend of estimated H derived from water in this figure is based on the isotopic equilibrium with the labeled water instead of an observed derivation of H from water in vivo into microbial biosynthesis. Again, as previous comment, showing the measured $_2$H abundances over the length of experimentation could provide insights to clarify these points.*

> Answer: We agree with the referee who pointed a key issue of the state of $H_{dfw}$ in the soil. We considered it as non-exchangeable hydrogen, either organic or inorganic. You mention the possible occurrence of water in the so-called inorganic HNE. We demonstrated the formation of organic HNE through the incorporation into the lipid fraction (table 3), and the linearity of this pool with the varied and/or nil amounts of added C (fig. 3 and fig. 4). The carbon dependency of hydrogen derived from water confirmed that the measured $H_{dfw}$ (fig. 2) is involved in biological reactions. However, inorganic HNE is not excluded in the measurement of $H_{dfw}$ as discussed in section 4.3. At the moment of the $\delta^2H$ measurement, non-organic H (which might be in the state of inorganic hydroxyl, hydrated ions, water in different states) was estimated from total H measurement of the dry soil as varying from 0 mg/g (podzol) to 6 mg/g (leptosol).
>
> You point out in your comment on Table 3 the difference between the bulk soil and lipids $H_{dfw}$ in one of the soil. In the most clayey soil, the inorganic $H_{dfw}$ pool is in an amount accounting for less than 1/40 of the amount of water-H. In any case, this pool appears very slowly exchangeable with "free water" (the magnitude of kinetic constant if any is in the range of weeks to months in situ at 28°C).
>
> Inorganic $H_{dfw}$ is discussed in detail in section 4.3. The assessment of presence-absence of a water contribution requires additional experiments, for instance warming the soil at different temperature levels. It would take the risk to be too simplistic in such a medium, e.g., that includes dynamic formation/destruction of poorly crystalline minerals, hydrated minerals, smectites saturated with hydrated calcium ions etc. It would require long methodological discussions, and we preferred in this paper the pragmatic estimate of HNE/HE in realistic conditions (ambient temperature and moisture).
>
> Action: We clarified the section 4. lines 347-355 p.10 and line 390-393 p.11. We also replaced the term "HNE", which is here conceptual and might be confusing" by "$H_{dfw}$", which corresponds exactly to the measurement.

*18. Table 3. One more noticeable thing in the table is that the results between H (% Hdfw) and C (% Cdfm) are quite consistent in lipids, which do not have exchangeable hydrogen. In the other hand, the proportion of hydrogen derived from the labeled source did not behave similarly in the bulk soil (with exchangeable H). A clear explanation on the treatment of exchangeable hydrogen can convince the reader on a differential isotopic routing of H and C.*

Answer: see comment 17

Calculation of water recycling:

See supplementary material, section 2.4. for equations references.

To calculate the incorporation of the water hydrogen coming from the mineralisation of the added molecule (recycling), we assume that the labeled molecule is completely mineralised in water. The resulting isotopic composition of water in experiment 1($A_{w2}$) can be calculated from the isotopic composition of the labelled molecule as follow:

$$A_{w2} = (A_m * H_m)/H_w \tag{SI9}$$

Then, the amount of non-exchangeable hydrogen that can be derived from this water ($H_{dfw2}$) can be calculated using the value $H_{dfw}$ calculated in equation (3) :

$$H_{dfw2} = (A_{w2} - A_{tot\_0})/ (A_m - A_{tot\_0}) * H_{dfw} \tag{4}$$

The proportion of deuterium derived from the molecule but incorporated in the soil by the water is given by ($H_{dfw2}/H_{dfm}$)*100 where $H_{dfm}$ is calculated in equation (2).

δ¹³C and δ²H results of the incubation samples:

[Figure]

Figure S4.1: Cambisol $^{13}$C and $^2$H isotopic variation  a. δ$^2$H variation through time of the bulk soil that received labeled glucose, phenylalanine, isoleucine and palmitic acid and unlabeled samples. b. δ$^2$H variation through time of bulk soil that received labeled water and unlabeled samples. c. δ $^{13}$C variation through time of the bulk soil that received labeled glucose, phenylalanine, isoleucine and palmitic acid and unlabeled samples. Standard deviations are less than 3 ‰ for unlabeled samples.

[Figure]

Figure S4.2: Podzol $^{13}$C and $^{2}$H isotopic variation a. $\delta^{2}$H variation through time of the bulk soil that received labeled glucose, phenylalanine, isoleucine and palmitic acid and unlabeled samples. b. $\delta^{2}$H variation through time of bulk soil that received labeled water and unlabeled samples. c. $\delta$ $^{13}$C variation through time of the bulk soil that received labeled glucose, phenylalanine, isoleucine and palmitic acid and unlabeled samples. Standard deviations are less than 3 ‰ for unlabeled samples.

[Figure]

Figure S4.3: Leptosol $^{13}$C and $^{2}$H isotopic variation  a. $\delta^{2}$H variation through time of the bulk soil that received labeled glucose, phenylalanine, isoleucine and palmitic acid and unlabeled samples. b. $\delta^{2}$H variation through time of bulk soil that received labeled water and unlabeled samples. c. $\delta$ $^{13}$C variation through time of the bulk soil that received labeled glucose, phenylalanine, isoleucine and palmitic acid and unlabeled samples. Standard deviations are less than 3 ‰ for unlabeled samples.

---

## Author Comment (AC2) · 3 Nov 2016

We thank the referee for the relevant comments which allowed us to improve the first version of our paper.

The referee comments are reported below with the answers we provided for them.

**Anonymous referee#2**

*This is an important paper in the context of understanding of hydrogen dynamics in soil organic matter. The paper is well written and sound.*

1. *I believe the backline story about Tritium detracts from the main study research outcomes as the context of tritium remains only touched upon, as we get no real concrete data about the concerns raised line 64/65. There is mention of enhancement tritium entering the environment due to historic bomb-testing but no mention that tritium is a radioactive form of H (half-life 12.3 years), unlike 2H and 1H who are stable isotopes, so will over time will dissipate and has done already decline since the bomb-14C peak. The authors should either reduce the tritium context or make it more quantitative.*

    Answer: We decided to reduce the tritium context in the text but we want to keep in readers mind that such a study on hydrogen dynamics in soil organic matter could be used for the prediction of tritium fate.

    Action: We reorganized the paragraph in the introduction about tritium in the line 60 p. 2.

2. *The authors may wish to comment on the potential of water in the air (different isotopic H signature) to enter the experimental jars and when opening them to prevent anaerobic conditions occurring in the jar.*

    Answer: By taking the value of the saturation vapor pressure at 28°C (28 g/m$^3$), the amount of water contained in the headspace jar (0.17 dm$^3$) was 4.8 mg. The proportion of labeled water vapor lost by the renewal of incubation jar headspace was estimated at 0.7 % the first months and 2 % at one year. The impact of the atmosphere renewal on the isotopic composition was therefore neglected.

    Action: We provided information in section 2.2.3, p. 4/5

3. *Tremendously (line 229) maybe find another word to describe the vary large increase.*

    Action: We removed the word that was not appropriated in this case.

4. *For Figure 2 and 3 the scale on the y-axis between the various soil types are different, which makes immediate comparisons difficult. If, the authors want to*

*retain this, maybe indicate in the legends of Figure 2 and 3 that this is the case
"note, the scale of Y-axis varies between the subfigures'*

Action: We changed the legend of figure 2 and 3 in this way.

---

## Referee Report (RR1)

Review of BG-2016-317 by Paul et al. (after major revision)

In the present version, the authors have done a good job in addressing the reviewers' comments and they provide enough reasoning for most of the comments from the previous review. The new figures presented in the Supplementary material (S4) help much to understand these experiments. The manuscript has been improved but I still have two comments that need some follow-up.

1. Hydrogen exchangeability – the authors equilibrated the samples with lab atmosphere for 2 hours. Is this enough? Could this statement be supported?

2. Sources of hydrogen and carbon – the definition of '% of C and H derived from the molecule' in the figures and the text is confusing to me. I think the authors mean the proportion of the remaining added molecule in the soil – but this can be misunderstood with the C and H amount incorporated into soil microbial biomass. I suggest that the authors clarify this point along the manuscript.

Technical comments
Line 258 – Eqs. (6) and (7)?

---

## Author Response (AR2)

Review of BG-2016-317 by Paul et al. (after major revision)

We sincerely thank the referees for their comments.

*In the present version, the authors have done a good job in addressing the reviewers' comments and they provide enough reasoning for most of the comments from the previous review. The new figures presented in the Supplementary material (S4) help much to understand these experiments. The manuscript has been improved but I still have two comments that need some follow-up.*

*1. Hydrogen exchangeability – the authors equilibrated the samples with lab atmosphere for 2 hours. Is this enough? Could this statement be supported?*

>   Answer: The time for exchange is usually very quick as shown by Schimmelmann et al. (1999) or Sessions et al. (1999) but it is true that exchange can be completed after few days at room temperature (Wassenaar and Hobson, 2000, 2003). However most of exchange occurred in few minutes (Wassenaar and Hobson, 2000). In the case of exchange of the dry soil with the atmosphere and the definition we provided for non-exchangeable hydrogen, 2 hours was enough to eliminate the contribution of exchangeable hydrogen in the final isotopic calculation and to deal with the objective of the study.

*2. Sources of hydrogen and carbon – the definition of '% of C and H derived from the molecule' in the figures and the text is confusing to me. I think the authors mean the proportion of the remaining added molecule in the soil – but this can be misunderstood with the C and H amount incorporated into soil microbial biomass. I suggest that the authors clarify this point along the manuscript.*

>   Answer: The molecule is degraded very quickly the first days of incubation. We do not trace the remaining molecule but the remaining C and H from the molecule.
>   The amount of C and H remained from the molecule in the total soil are expressed in percent relative to the amount of C and H initially added as molecule.
>   We added details in the captions of the figures 1 and 2 and in lines 261-263; line 283 to avoid confusion.

*Technical comments Line 258 – Eqs. (6) and (7)?*

>   Answer: We changed Eqs (6) and (7) by (1) and (2).

[revised manuscript text omitted]

Fig. 1

[Figure]

[Figure]

610

Fig.3

[Figure]

[Figure]